# An Exploratory Study of Helping Undergraduate Students Solve Literature Review Problems Using Litstudy and NLP

**Gary K. W. Wong and Simon Y. K. Li \***

Faculty of Education, The University of Hong Kong, Hong Kong 999077, China; wongkwg@hku.hk
\* Correspondence: simonykli2004@gmail.com

**Abstract:** (1) Many undergraduate students struggle to produce a good literature review in their dissertations, as they are not experienced, do not have sufficient time, and do not have the required skills to articulate information. (2) Subsequently, we deployed Litstudy and NLP tools and developed a recommendation system to analyze articles in an academic database to help the students produce literature reviews. (3) The recommendation system successfully performed three levels of analysis. The elementary-level analysis provided demographic statistical analysis to the students, helping them understand the background information of the selected articles they would review. The intermediate-level analysis provided visualization of citations in network graphs for the students to understand the relationships of the articles' authors, regions, and institutes so that the flow of ideas, development, and similarity of the selected articles can be better analyzed. The advanced level of analysis provided topic modeling functions for the students to understand the high-level themes of the selected articles to improve productivity as they read through them and simultaneously boost their creativity. (4) The three levels of analysis successfully analyzed the selected articles to provide innovative results and triggered the students to handle literature reviews in a new way. Further enhancement opportunities were identified in integrating the NLP technologies with large language models to facilitate the generation of research ideas/insights. This would be an exciting opportunity to have AI/NLP integrated to help the students with their research.

**Keywords:** literature review; topic modeling; NLP; litstudy; AI; LLM



## 1. Introduction

Most students struggled with synthesizing, critiquing, and explaining literature in their writing and, instead, primarily focused on summarizing others' findings and interpretations. Additionally, insufficient knowledge, time constraints, and negligent supervision hindered the student's ability to write effective literature reviews. When researching a novel area, finding answers to broad questions related to the overall topic can become challenging due to the large volume of literature available. Searching for a researchable problem is often difficult for undergraduate students and seasoned researchers. Addressing these problems could improve student motivation and positively impact their writing experience.

Therefore, it is imperative to develop an innovative approach to assist students in identifying a research gap at the initial stage of a project, whereby teachers can spend more time supporting students carrying out a research project. This Teaching and Development Grant (TDG) project fund supported the development of an artificial intelligence-enabled recommendation (AIR) system using advanced data mining and natural language processing (NLP) [1] techniques, which help to understand human languages with computer algorithms to support the final year project (FYP) students in their project preparation and ideation for research topics. A one-year AIR system development project was then proposed.

We developed a pilot academic information recommendation system to ease the above problems. We have utilized Litstudy, a Python library that allows users to quickly analyze scientific papers by applying visualization methods, to develop a recommendation

system for literature reviews. This paper will present the results of an experiment using the academic information recommendation system designed and developed to handle the problems mentioned above. At the same time, NLP methods were also applied in our experiment for textual analysis. The library Litstudy simplifies literature reviews through easy-to-understand and stimulating graphical analysis to help the students better understand literature reviews. This AIR system was piloted with the BSc (Information Management) (BSIM) and BASc (Social Data Sciences) (BSDS) programs. It is expected that this AIR system, delivering a prototype of an ideation recommendation system, will have the potential to apply to all future research projects of students at both undergraduate and postgraduate levels.

## 2. Theoretical Background

### 2.1. What Is a Literature Review

A literature review critically analyzes written materials about a particular topic, typically conducted in academic fields such as science and social sciences. It serves several purposes, including providing background information, highlighting gaps or inconsistencies in existing knowledge, summarizing evidence, evaluating arguments, synthesizing ideas, formulating research questions, and determining directions for further investigation [2]. It is essential for writing an effective thesis. Various definitions and classifications of "literature reviews" exist in the literature. Such descriptions include reporting on primary or original studies and describing, summarizing, evaluating, clarifying, and integrating their content [3]. Succinctly, a literature review is a standalone article connecting previous and current findings on a particular subject [4]. It is essential in structuring research topics and providing links between various studies [5]. Writing a good literature review requires attention to covering all areas and ensuring accuracy. Literature reviews facilitate delimitating research questions, generating hypotheses, and determining appropriate research designs. Poorly written literature reviews lead to incorrect analyses of the results of prior studies and misinterpretations of research questions. On the positive side, strong literature reviews enhance analytical papers by enabling readers to appreciate existing research issues better, understand the study context, compare different perspectives and theories, and apply ideas from prior investigators effectively.

A literature review entails critically evaluating written works published on a particular topic or subject area. It involves a comprehensive search of relevant publications, analyzing their quality and scientific rigor, and synthesizing findings to create a coherent narrative summarizing existing knowledge about the topic under consideration. Moreover, it aims to identify key themes, patterns, gaps, or conflicts in the literature and serves as an essential foundation for conducting empirical research, forming hypotheses, or developing theories. Literature reviews can take various forms and serve several purposes, including assessment of research progress, validation of research questions, justification of experimental designs, or refinement of conceptual frameworks. They are commonly included in academic papers, reports, dissertations, grant proposals, or research plans to inform stakeholders about the state of knowledge in the field and guide decision-making [6].

Literature reviews usually include a critical analysis of published sources and research findings relevant to a particular area of study or investigation. Its primary purpose is to summarize and synthesize existing knowledge on a specific topic, highlight important themes and trends, identify gaps in existing research, and evaluate the quality and reliability of the information available. By thoroughly reviewing published materials, authors aim to provide context, background information, and support for their own research questions, hypotheses, and conclusions. Literature reviews are commonly found in academic journals, books, dissertations, and conference proceedings. They serve as a foundation for new empirical investigations or theoretical developments that build upon the existing body of knowledge in a field of inquiry [7].

### 2.2. Literature Review Problems

Students face several challenges when writing the literature review section of their thesis. These challenges include limited time and resources, a lack of experience, difficulty synthesizing information, etc. This urges the need for a study specifically with students in mind to explore the extent to which challenges associated with academic writing influence the process of writing literature reviews. These challenges will be reviewed individually in the following paragraphs.

### 2.2.1. Lack of Knowledge of Writing an Effective Literature Review

Most students lack knowledge regarding the significance of the literature review. They believe it solely involves summarizing existing research while failing to grasp its purpose in obtaining a deeper understanding of the topic and studying the methods employed in past exploration. Almost all students underlined the importance of neutrality when authoring the literature review. They emphasized utilizing multiple perspectives effectively and determining where each author stands, enabling readers to assess carefully. They also preferred authors who neither mentioned nor presented conflicting points but expressed genuine thoughts [2]. On the contrary, some students retell the opinions found in literature reviews, ignoring creativity. Other problems include utilizing updated references, including high-caliber ones, in literature reviews, which is crucial for upholding exactness. They mainly concentrated on extracting relevant content and accepted that they might miss appropriate records if gathering finely written reports required extra time. Despite these concerns, few students felt bound to be careful while assembling papers. These discoveries affirm the finding of gaps among students identified elsewhere, indicating certain regions need reinforcement to help learners understand what makes an excellent literature review besides offering hands-on exercises in creating one.

### 2.2.2. Time for Completing Theses and Publications

Most students admitted to facing challenges due to insufficient time, resulting in lower standards for their literature review sections. Many expressed concerns over the limited time to conduct their thesis work. This constraint may lead to lower quality in the literature review section of their theses [2]. One individual even admitted that they prioritized finishing the thesis quickly so they could start their own business instead of ensuring the quality of their literature review. Some other students seemed indifferent toward the quality of their literature review. Conversely, another student placed significant importance on crafting an exceptional literature review to ensure success in their career path. Although most students acknowledged the importance of incorporating up-to-date and high-caliber articles in their review studies, many still failed to adhere to this practice due to challenges in accessing quality sources. Additionally, advisors' scrutiny regarding article selection appeared insufficient to motivate better practices [2]. A disconnect exists between desired quality expectations and actual behaviors exhibited during the literary search. Enhanced training in effective research methodologies seems necessary to close this gap, perhaps through direct involvement with experienced academics. Notwithstanding, there were still students determined to produce the highest quality work despite potential difficulties. Most students, though, confessed not prioritizing high-quality, up-to-date material when discussing article choice. Several factors seemed responsible, ranging from a focus on simply completing tasks to an apparent disinterest in academic careers. Additionally, supervisor scrutiny appeared minimal, discouraging rigorous article selection. These results suggest that support may be necessary to encourage higher standards in thesis creation.

### 2.2.3. Supervisors and Professors' Role in Writing the Literature Review

Most students felt that their instructors' teaching of research methods did not adequately emphasize writing an effective literature review, potentially leading to subpar outputs. On the other hand, some students reported receiving minor edits and suggestions

from their supervisors, predominantly concerning grammar, punctuation, reference accuracy, and structural elements in their literature review. However, some students reported that supervisors often provided minimal input unless they were interested in the research outcomes [2]. Furthermore, most students believed their supervisors and committees spent little time evaluating their literature review compared to other thesis sections. Interestingly, a few students shared instances where their supervisors encouraged copying content directly from other papers or relying on paid editing services. Few students appreciated their supervisors' thorough critique and advice on improving their literature review. These varying experiences indicate diverse levels of guidance available to students and underline the significance of developing suitable evaluation techniques for mentoring literature review writing abilities [2].

### 2.3. Natural Language Processing (NLP)

Natural Language Processing (NLP) [1] is a subfield of artificial intelligence and computer science that focuses on the interaction between computers and human languages. It involves developing algorithms, software, and models to enable computers to understand, interpret, analyze, generate, and respond to human languages in various contexts, such as text analysis, topic modeling, speech recognition, machine translation, question answering, and chatbots. NLP techniques are used in many applications, including search engines, virtual assistants, customer service bots, automated writing tools, and language learning platforms.

### 2.4. Topic Modeling

Topic modeling [8] is a technique used in Natural Language Processing (NLP) [1] to discover patterns of topics in extensive collections of textual data. Topic modeling is used to identify groups of words that tend to occur together within one collection and represent coherent thematic topics based on a probabilistic topic model [9]. These topics are derived from underlying statistical dependencies among words and document structure. Topic Modeling entails identifying hidden "topics" in a text body. Such topics represent unique themes or ideas within the text, consisting of multiple associated words. These topics are modeled automatically, solely based on the sequential order in which they appear in the input corpus. Previous iterations of traditional latent semantic indexing techniques relied upon the manual specification of codes, which were later compared against indexing results via success metrics [10].

Topic Models employ latent structure discovery tools to identify hidden thematic groupings within extensive collections of words called documents. They enable users to explore complex relationships among these topics, revealing underlying patterns in word co-occurrences. By analyzing word vectors, topic models cluster terms according to their shared contexts, generating compact, high-level abstract representations that allow humans or machines to swiftly grasp salient information or navigate massive databases without requiring exhaustive keyword searches [11]. Important algorithms for topic model development include Latent Dirichlet Allocation (LDA) [12] and Latent Semantic Analysis (LSA) [13], each relying on Bayesian inference and tensor decomposition principles [14]. Their practical applications range from enterprise document management and knowledge discovery to text corpus analysis for digital humanities scholars seeking to expose historical changes in discourse styles or literary movements' evolutionary trajectories using topic modeling methodologies tailored to their domain-specific challenges.

Topic modeling is a powerful tool that gives us insights into how different concepts interrelate in a given corpus or body of text [15]. In principle, topic models try to identify the key topics discussed across multiple documents and assign each word present in those documents to a particular case. By grouping similar words under an umbrella topic, it becomes relatively easier to understand how certain concepts are linked, making it simpler to determine how these interrelated topics have temporally evolved. Topic modeling has many applications in NLP. One of the most common use cases is analyzing an extensive

corpus/collection of news articles and finding out topic prevalence trends based on a date range. In another sense, it helps recommendation systems generate appropriate items for user interest by analyzing user profiles generated from past interactions/clickstreams, etc., data collected during logins to websites/opening apps, or other digital touchpoints, further refined by detecting seasonal/pattern changes, weather conditions, and geographical locations, etc.

*2.5. Recommendation System*

The recommendation system is not a new concept in educational technology. It is a software application "capable of presenting a user a suggestion for an object, obtained based on his previous preferences and the preferences of a community, which has likings and opinions similar to him" [16]. It aims to reduce our overload of big data and massive information by offering selected access to useful information for any specific domain [17]. However, old recommendation systems typically focus on factual data (e.g., citation, key-word matching), "Like" given by human users based on their recommendation on any particular object (e.g., journal articles), or even using ontology network analysis to select papers based on the user's profile of interest, instead of focusing on the research gaps that students can look into [16,18]. With the help of advanced data sciences and AI technologies, the latest recommendation system techniques focus more on the semantics and meanings through advanced data mining or AI technologies, such as natural language processing (NLP) and sentiment analysis [19,20].

## 3. Methods

To minimize the time for students to identify a feasible research topic, NLP technologies could be applied to help analyze the big text-based data among all existing research papers and reports and sort out some viable topics with a list of recommended papers for their literature review. With these recommendations, students can focus on developing research questions and designs with their supervisors instead of having no clue about how to create their research project based on their interests. Moreover, adopting advanced text mining and NLP technologies would enable more efficient identification of promising research topics, facilitate speedier literature searches, and encourage greater collaboration with advisors to devise research agendas tailored to student aspirations. In essence, deploying cutting-edge technology alleviates common obstacles associated with scoping literature reviews, enabling deeper discussions on crucial aspects of a research proposal. Thus, an artificial intelligence-enabled recommendation (AIR) system is proposed to be built to help students in FYP preparation.

To the best of our knowledge, a recommendation system for undergraduate research students to identify the current research trend/gap has not been a widely adopted concept or practice by educational technologists. While AI in education is a new emerging field, AI techniques with advanced data/text mining are becoming more mature. It is now the right time to explore how AI and data/text mining can help support students' learning in their FYP. The proposed system starts with importing the textual content of academic articles. The text materials will then be analyzed by NLP technologies to provide insights to students to ease their literature review tasks.

In our experiment, we built a recommendation system using Python 3.9.12. A Python package named Litstudy [21] was adopted and modified in our system with another package called mljar-mercury 1.1.5 so that the graphical outputs could be transformed into a web format to be displayed using a web browser. A live connection to Scopus was implemented in the AIR system so that the articles' metadata could be retrieved there. To ensure the quality of the articles, if they cannot be found in Scopus, they will not be included in the analysis by the AIR system.

The users search the academic articles from SpringerLink with key terms as the search criteria (Figure 1). The articles matching the search criteria will be listed, and the users must download them as a CSV file containing the articles' metadata. The CSV file would

then be uploaded into the AIR system for analysis (Figure 2). In other words, only the articles found on SpringerLink and Scopus will be included in the analysis conducted by our experiment. Three levels (elementary, intermediate, and advanced) of analysis were provided to address the literature review problems identified above: limited time and resources, lack of experience, and difficulty synthesizing information.

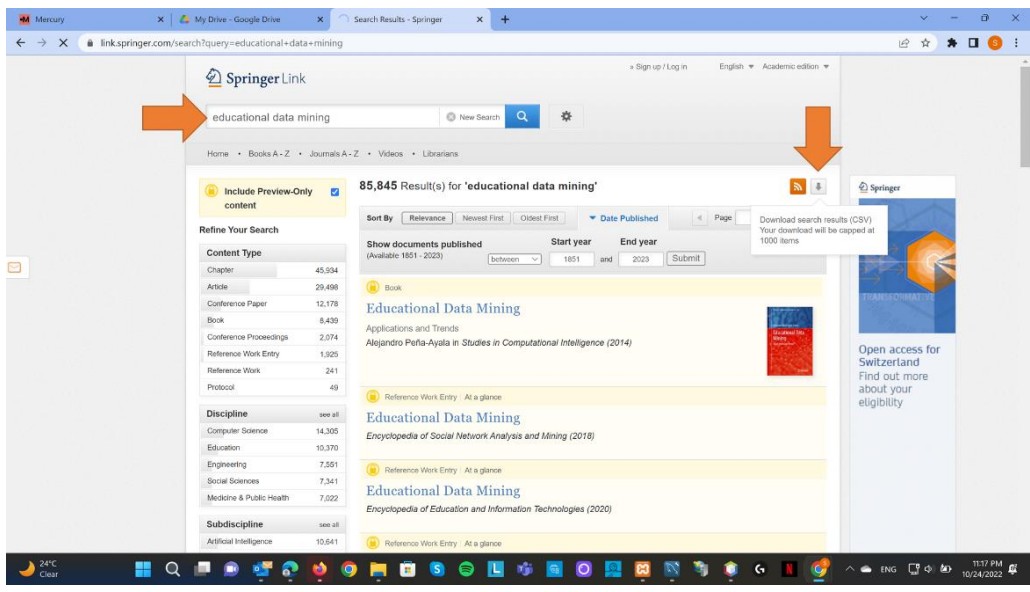

**Figure 1.** Search articles with SpringerLink.

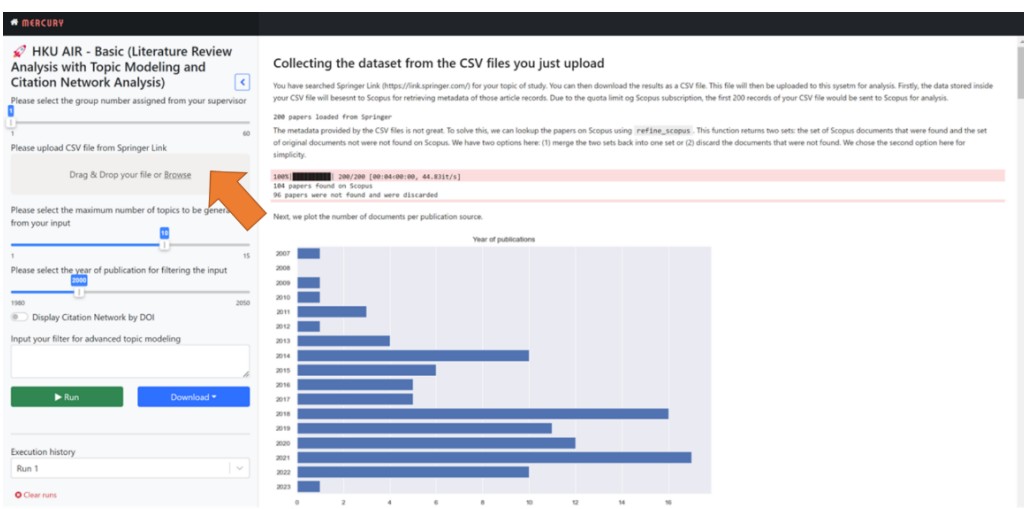

**Figure 2.** AIR screen for uploading CSV files.

Figure 3 summarizes the overall flow of our experiment, showing the features, steps, and functions of the three levels of analysis. It is a case of human-in-the-loop computation [22]. Human participation should not be removed. Humans should be included in the model development cycle of machine learning [23] to achieve optimized outcomes. Large language models (LLM) can be a good candidate for improving recommendation systems [24]. LLMs can be an open-ended recommendation system [25]. However, the recommendations can be overwhelming and not precise enough. Therefore, we may need more accurate input to limit the recommendation output. Potential enhancements, primarily on the new applications of the key terms extracted from the topic models (Figure 3) in the generative AI areas (region shaded in yellow in Figure 4) to inject new idea generation, are also shown here, which will be discussed in the next section.

```
Topic 1: ['research', 'review', 'security', 'journal', 'cultural']
Topic 2: ['mnes', 'fdi', 'countries', 'government', 'acquisition']
Topic 3: ['supply', 'supply_chain', 'chain', 'supplier', 'sustainable']
Topic 4: ['firms', 'market', 'competition', 'product', 'platform']
Topic 5: ['retail', 'channel', 'internet', 'entry', 'manufacturing']
Topic 6: ['port', 'shipping', 'reform', 'sector', 'global']
Topic 7: ['quality', 'banks', 'disclosure', 'risk', 'transparency']
Topic 8: ['negotiation', 'web', 'service', 'component', 'architecture']
Topic 9: ['tax', 'trade', 'stock', 'price', 'exchange']
Topic 10: ['online', 'mobile', 'trust', 'commerce', 'service']
```

**Figure 3.** Examples of topic models.

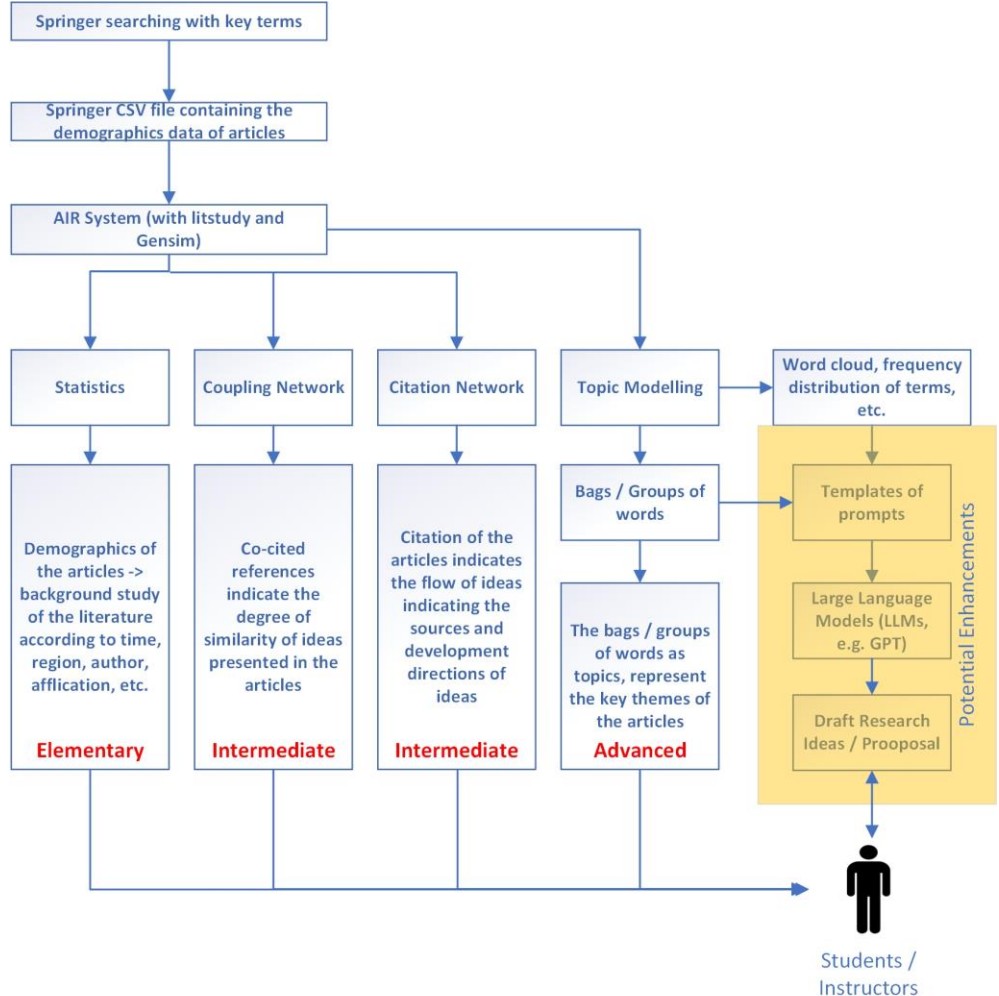

**Figure 4.** The overview of the experiment design.

## 4. Results

### 4.1. Elementary-Level Analysis

The elementary level of analysis refers to the demographic statistical analysis of the articles, which includes the following items:

- The number of articles published each year (Figure 5): This information is an obvious indicator of the popularity of the research topic across different periods. The trend may look like a product life cycle. If it has been popular for several years, it would be better to avoid it if new and original research is preferred. On the other hand, it would be good if a simple general review of the mature research articles was chosen.

- The number of documents by author affiliation (Figure 6): The number of documents indicates the potential capability of a particular academic institute. You may contact those institutes with a higher number of documents by author affiliation so that you may obtain more relevant information.
- The number of documents published per author (Figure 7): The number of documents indicates the potential capability of a particular author. This information helps students make more intelligent choices for authors to pursue in their literature selection. More productive authors may indicate that their research areas are popular. Whether you will avoid those areas or not depends on your preferences. Probably, those authors are the pillars of those areas. Based on the authors' names, it may also be possible to guess which countries may be interested in the topics of your articles.
- The number of documents by language (Figure 8) indicates the dominant language used in the research. Based on this information, the students may try publishing academic papers in other dominant languages.
- The number of authors per document (Figure 9) indicates the human resources used in the research. Based on this information, the students can also plan their project teams to proceed with their research.
- The number of documents by country of author affiliation (Figure 10): The number of documents by country indicates the dominant country in the research. The students may plan their future research in less dominant countries to avoid head-to-head competition by borrowing knowledge from those countries. The students may even prepare for international collaboration among dominant and less dominant countries to facilitate knowledge transfer.
- The number of documents by continent of author affiliation (Figure 11): This number indicates the dominant continent(s) in the research. The students may plan their future research in less dominant areas to avoid head-to-head competition by borrowing knowledge from those dominant areas.

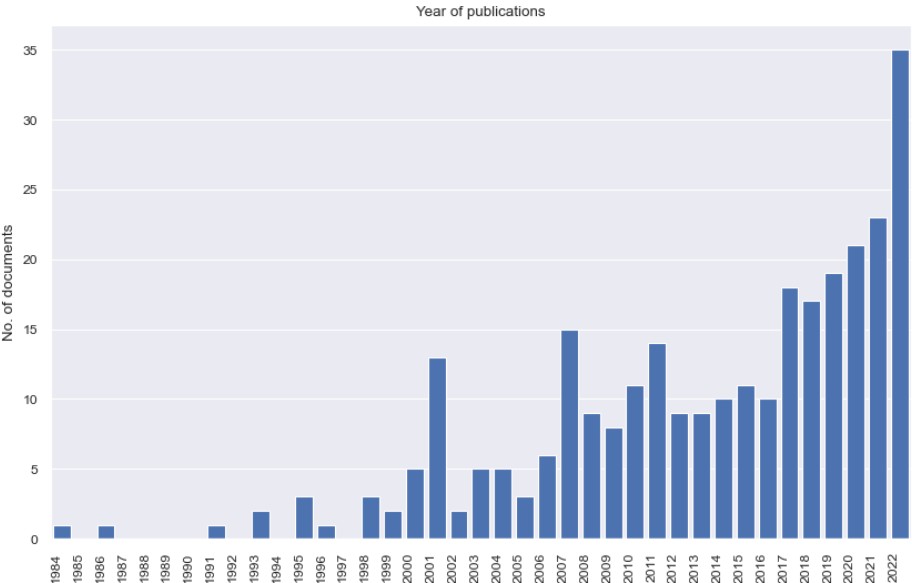

**Figure 5.** The number of articles published each year.

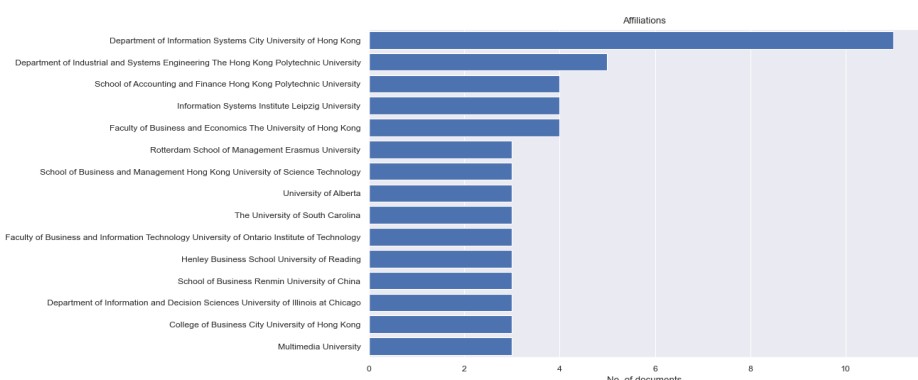

**Figure 6.** The number of documents by author affiliation.

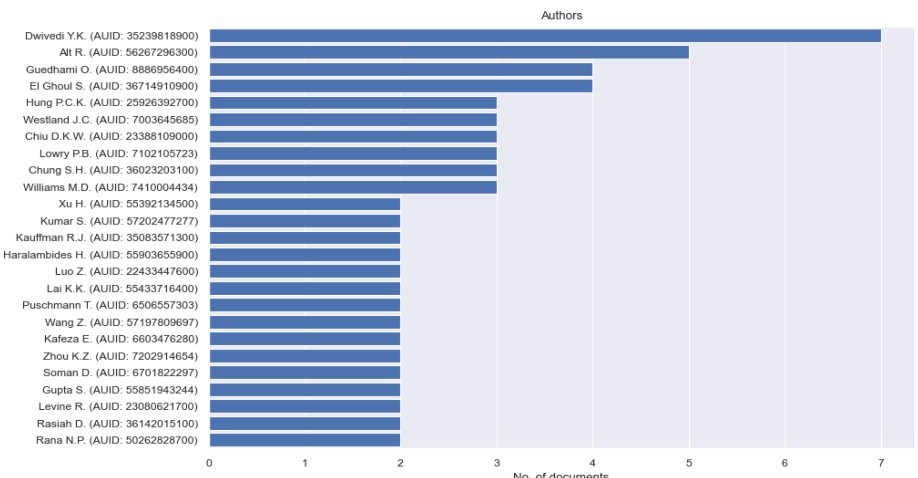

**Figure 7.** The number of documents published per author.

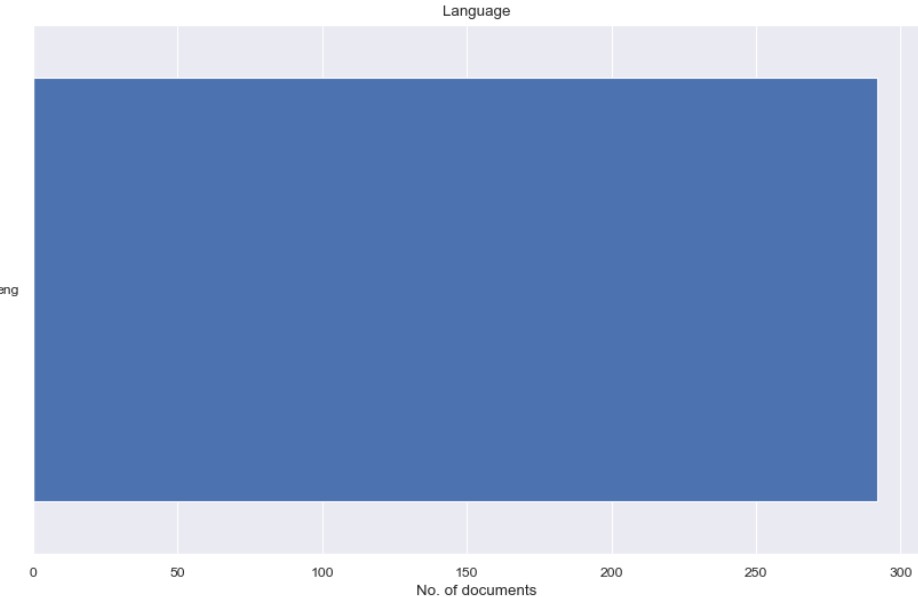

**Figure 8.** The number of documents by language.

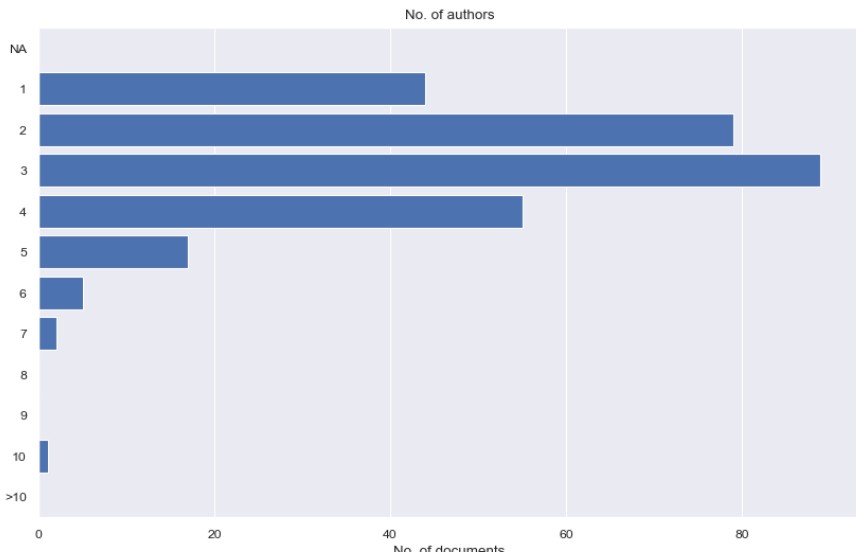

**Figure 9.** The number of authors per document.

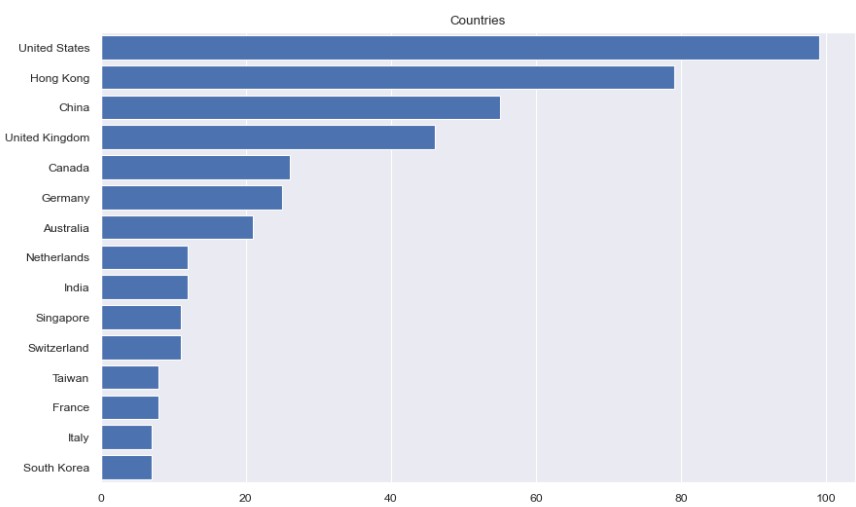

**Figure 10.** The number of documents by country/region of author affiliation.

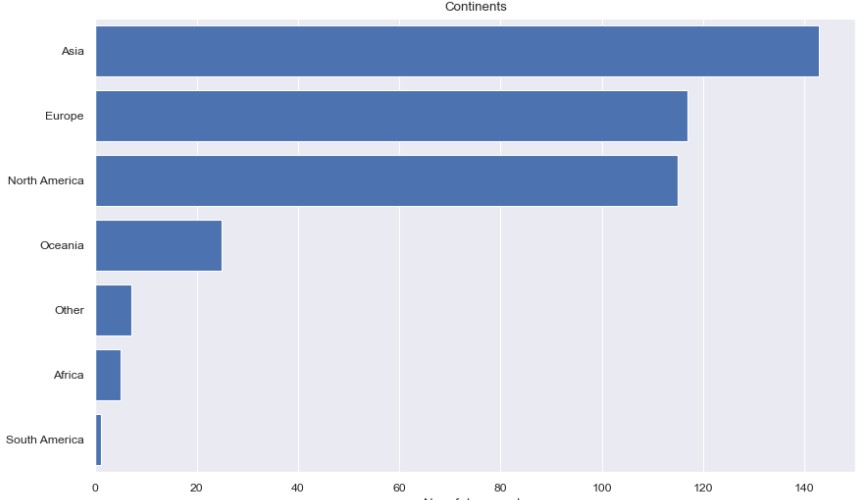

**Figure 11.** The number of documents by continent of author affiliation.

### 4.2. Intermediate-Level Analysis

The intermediate level of analysis includes the coupling network and the citation network. This recommendation system analyzes the articles' metadata using Litstudy to produce coupling and citation networks to visualize the connections in the literature. A coupling network (Figure 12) is an undirected graph where nodes indicate documents and edge weights store the bibliographic coupling strength. This strength measures how similar two documents view related work. It is calculated as the number of shared references between two documents. The coupling network, which may be the most useful for recommending articles for the student to select to read, shows the shared references between two articles in their reference sections. Sharing references between any two articles means the articles may be highly likely based on some similarly referenced ideas. In other words, they may be highly similar to each other from the perspective of content.

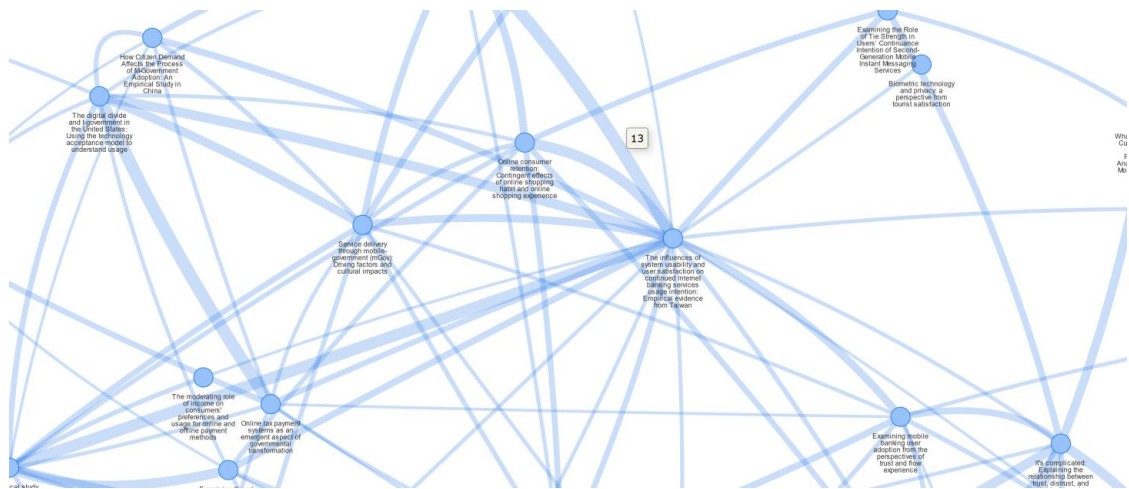

**Figure 12.** A sample Coupling Network.

Therefore, the number of shared references between two articles, represented visually by the link thickness, can be used to recommend which pair of articles to read. The higher the number of shared references (or the thicker the link), the higher the similarity between the two articles would result. Suppose the students want to have a comprehensive review of the current literature. In that case, the cluster of articles with higher numbers of shared references may be a good choice since they may represent the "norm" of the discussion of the current topics. However, suppose the students want to have some novel discussion or content or identify a potential research gap. In that case, some clusters of articles with lower numbers of shared references, or even some clusters with no direct connections, may be better choices since those different shared references probably indicate a higher possibility that the ideas inside those articles may differ. It depends on the objectives of the students' theses; they can select which clusters of articles to review first if limited time is available. However, a balanced literature review should generally contain a comprehensive review of frequently discussed topics and some novel topics. In realistic situations, the time available for a literature review may not be enough. If the students can make informed decisions based on whether the clusters of articles contain general or novel information, it would be beneficial to make a wiser choice.

The citation network is another visualization from the intermediate-level analysis (Figure 13). It can also help trace the flow of ideas developed in the clusters of articles. It indicates the origins of the ideas and their subsequent developments. The citation network can provide more advanced guidelines to the students if they want to investigate how the concepts they referenced were developed to find any potential enhancements by alternating factors, such as the progressive changes made by each subsequent article following the article that posted the original ideas. This way, potential research gaps may be identified by

suggesting alternations different from the initially developed enhancement. Each bubble is an article. The arrow means a citation between two articles. A bigger bubble represents a more important article since many other articles have cited it. Thus, that article can be regarded as the "source" of some vital ideas referenced in many other articles. As general advice, those articles represented by the bigger bubbles should be read first if limited time is available for the students since they contain the original ideas being referenced by many others. If time permits, those connected articles along the arrow may be included for reading at a later stage.

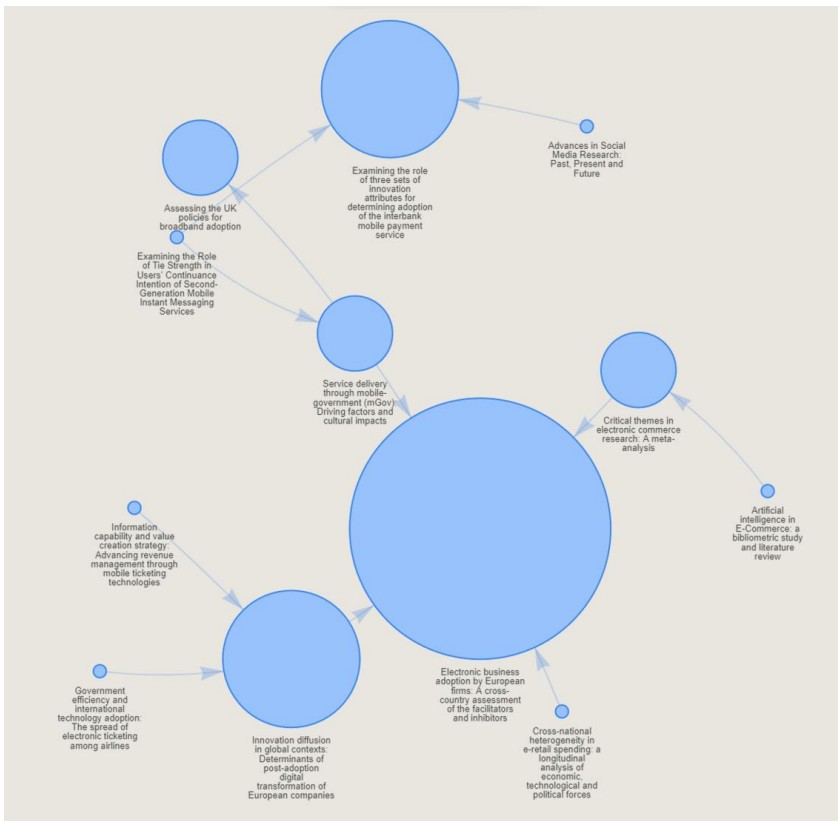

**Figure 13.** A sample Citation Network.

### 4.3. Advanced-Level Analysis

The advanced level of analysis includes natural language processing (NLP) techniques (a branch of text mining) such as term frequency analysis (Figures 14 and 15), word cloud (Figure 14), and topic modeling (Figures 14 and 16) with the package called gemsim [26]. NLP techniques are also a foundation for building intelligent systems such as artificial intelligence (AI) to understand human languages. This level of analysis deals with brainstorming ideas based on extracting some meaningful patterns from the abstracts of the articles. The frequent terms (Figure 17) of the documents are extracted. Based on the frequent terms sorted in order (Figure 16), it is easier to know the high-level ideas of the documents (as topics) represented in key terms and their groups/topics (Figure 14). Probably, time can be saved without reading all the documents to understand the "key themes" of the documents. Document or concept classification/selection can be made possible based on the key terms. The topics can also provide input to formulate (re)combinations of terms for a new search of articles using academic search engines. The collection of terms indicates a potential "topic." In other words, from the perspective of text mining or natural language modeling, a topic is a multi-dimensional model containing different terms (Figure 14) representing different dimensions (attributes or properties) of that topic. Bigger terms in the word cloud visualizing the topic models (Figure 15) are more important in representing key ideas. The topics represent the themes or terms that can express the main

ideas of the articles. The students can then spend less time extracting the key ideas from the articles without reading them all. The students can also create a story/narrative/research topic with those terms. Alternatively, the literature search can be restarted using and/or modifying those terms found in the topic models since they represent the key ideas of the articles selected in the analysis. This promotes creativity and innovation in research.

```
Topic 1: ['features', 'selection', 'dataset', 'performance', 'prediction']
Topic 2: ['recommendations', 'personalized', 'pedagogical', 'web', 'subjects']
Topic 3: ['big', 'analytics', 'literature', 'organization', 'edm']
Topic 4: ['machine', 'model', 'classifiers', 'algorithms', 'efficiency']
Topic 5: ['research', 'field', 'review', 'examination', 'methods']
Topic 6: ['school', 'dropout', 'characteristics', 'self', 'bayesian']
Topic 7: ['knowledge', 'activities', 'clustering', 'rules', 'patterns']
Topic 8: ['learners', 'teaching', 'materials', 'video', 'measure']
Topic 9: ['visualization', 'academic', 'analytics', 'prediction', 'information']
Topic 10: ['behavior', 'learners', 'clustering', 'moocs', 'styles']
```

**Figure 14.** Topics as multi-dimensional models.

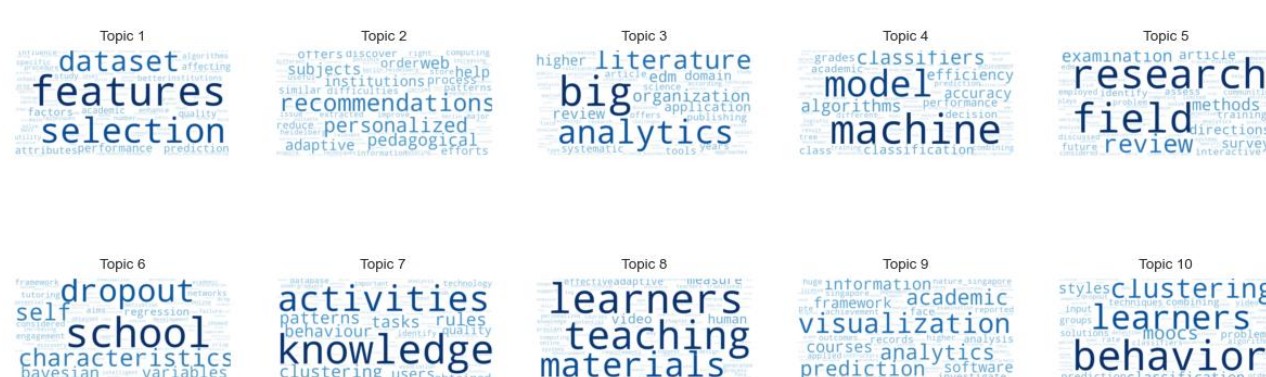

**Figure 15.** Topics visualized in word clouds.

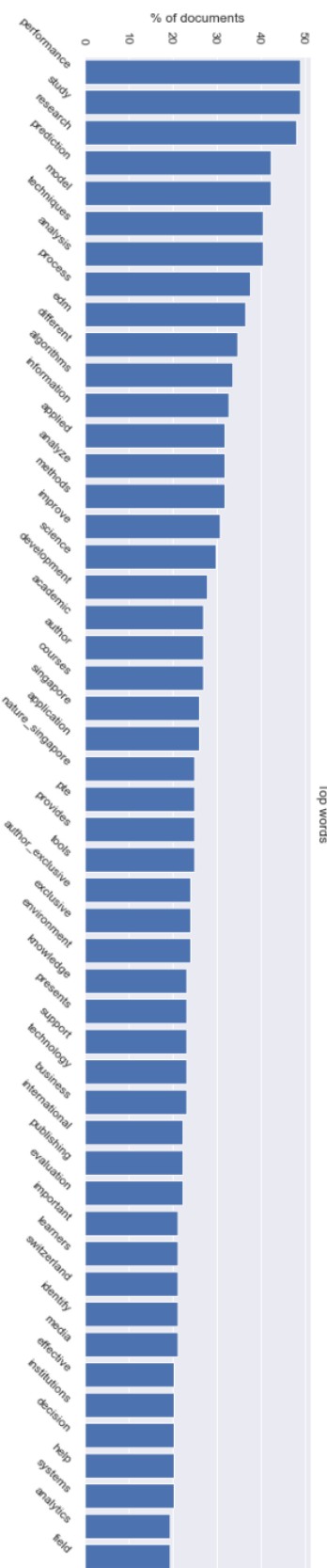

**Figure 16.** Frequent terms sorted in descending order.

| | count |
|---|---|
| **author_exclusive** | 25 |
| **berlin_heidelberg** | 6 |
| **business_media** | 8 |
| **decision_tree** | 9 |
| **intelligent_tutoring** | 5 |
| **international_publishing** | 19 |
| **licence_science** | 8 |
| **logistic_regression** | 5 |
| **media_llc** | 5 |
| **naive_bayes** | 5 |
| **nature_singapore** | 26 |
| **science_business** | 14 |

**Figure 17.** Frequent terms.

## 5. Limitations

The AIR system is based on the LitStudy package, which relies entirely on Scopus to obtain the literature metadata. This is the major limitation. Moreover, we just included the literature found at Springer. However, not all literature from Springer can be found on Scopus. Those articles not found in Scopus would not be included in our analysis. The authors of any articles would have the freedom to cite any articles that may not be found in Springer or Scopus. Unless we can have all the articles published accessible to our AIR system, this limitation may not be easily overcome. Subsequently, to improve the system's usability and make the AIR system applicable to all university study levels, we may need to modify the Litstudy package to look for information from open academic databases such as Google Scholar. However, this would be a massive redevelopment under the current situation unless additional resources could be available to the project team. Adding support for important article information from other academic databases, such as IEEE, ScienceDirect, etc., would be relatively easy as an interim improvement.

## 6. Contributions

In this project, we developed the AIR system to provide students and their FYP supervisors with some new tools to help both of them see literature reviews in some new ways at three levels of analysis (elementary, intermediate, and advanced). The Litstudy package was designed for interactive Python notebooks. We developed a web front end for the package using another Python package called Mercury. We modified the Litstudy package to redirect the outputs to files so that multiple users can access a web browser without installing those Python notebooks and packages, which would take considerable time for many students.

The elementary-level analysis raised the students' overall awareness of the demographics/environmental factors of the literature. However, this level of study is relatively straightforward. Most students might not consider those demographics or environmental factors before using the AIR system. They might pick up some articles when they encounter them without filtering due to a lack of time.

The significant contributions in the intermediate-level analysis are related to linking different directly or indirectly related articles together to form a network structure. In

addition to increasing the awareness of the student's ability to view literature in a connected network, we added our interpretation of those connections (in the Coupling and Citation Networks) of the articles based on our experiences to help the students select which articles to read by understanding how to make use of the connections to evaluate the relevance and importance of the literature. The Python package Litstudy did a good job; however, it lacks interpretation advice from the visualization means. This would be one of the missing pieces that most suffering students need. In other words, the students need consultation on comprehending the outputs produced by Litstudy.

Furthermore, we explained how to understand and utilize the NLP features of topic modeling at the advanced level of analysis. In addition to simple frequent term analysis, we helped the students view the topic models in terms of "topics" and their key terms to understand the most representative and trendy ideas from the literature selected for analysis. In this way, the students could have a glimpse into understanding whether the collection of literature fits the purposes of their research projects. The topic models contain key terms which are good candidate terms for any search engine. Modifying those terms and restarting the search process with modified/new search terms might be the next step to refine the search results. This is an iteration for the improvement of search results. Probably after several iterations, the students can find better literature that is more suitable for their research needs. Those iterations might also accidentally introduce helpful ideas for the students to refine or redefine their research questions.

## 7. Discussion and Further Development Opportunities

NLP has been an established practice for some time, yet there remains room for growth and improvement. Generative Artificial Intelligence (AI) [27] is among the latest advancements, focusing on generating textual content via advanced algorithm design. Despite their apparent opposition to the idea behind text mining, where the goal is to discover previously unknown patterns in existing data, these two concepts may work together. As we delve deeper into AI, it becomes increasingly apparent that the lines between various subfields within AI, NLP, and even humans are becoming blurred [28]. It is especially true when considering how seemingly disparate techniques such as text mining, generative AI, and human intelligence can complement each other. By embracing this interdisciplinary approach, researchers and developers stand poised to create groundbreaking tools capable of tackling even more significant challenges. Researchers can achieve powerful new ways of analyzing and understanding written communication by utilizing these methods together. One prominent example of this technology is GPT, a widely recognized tool known for its versatility and effectiveness in the media [29].

The acronym "GPT" [30] stands for "Generative Pre-trained Transformer," which was first released by OpenAI back in June 2019 as Version 1. Since then, four versions have been released, showcasing the rapid pace of progress in this cutting-edge technology. As a form of Generative AI [27] focused on natural language generation [31], GPT is crucial in enabling machines to process and produce human-like texts through advanced algorithms. In contrast to text mining, which centers around extracting useful information from unstructured data sets, GPT shines at handling open-ended queries and prompts given by users. Its capabilities include responding to questions, performing language translation, and completing sentences based on user input. These abilities resemble human traits in tackling unpredictable and adaptive situations [32]. The combination can lead to groundbreaking outcomes when integrated with NLP and text mining tools [33].

For example, a successful working prototype (Figure 18) was built to demonstrate the extraction of the terms of the topic models (shown in Figure 16) and building up a simple prompt such as "Produce a research proposal based on the following keywords: (list of some selected terms in the topic models)." Afterward, via application programming interfaces (APIs), the prompt was sent to a large language model (LLM) [34], such as Huggingface [35,36], etc. A research proposal was generated with some interesting content. The output from this prototype seemed promising, although a lot of fine-tuning would still

be needed, especially for template/prompt design [37], to generate more precise results along with information accuracy and relevance validation. It is important to stress that we do not want to replace the participation of the students and teachers in discussions on how to finish the literature review for their dissertation tasks. Instead, we aim to achieve AI with human-in-a-loop [23] capabilities.

```
print(chatbot.chat("Write a proposal based on the following key words: children, digital, devices, interest, iot"))
```

```
Proposal Title: Enhancing Children's Digital Interest through Interactive IoT Devices

Introduction:
With the rapid advancement of technology, there has been a significant increase in the use of digital devices by children. These devices not only provide enter
tainment but also play a crucial role in their cognitive development. However, most digital devices are passive tools that lack interactivity, which can cause
boredom among children. To address this issue, we propose a new approach using Internet of Things (IoT) devices that would enhance children's digital interest
by making them more interactive. This project seeks funding to research and develop these innovative IoT devices for children.

Project Objectives:
The primary objective of this project is to design and develop IoT devices that engage children and foster their creativity while providing educational value.
Our objectives include:

1. Identify common interests and preferences among children aged 6-10 years old.
2. Develop and test prototypes of IoT devices tailored to meet the identified interests and preferences.
3. Evaluate the effectiveness of the developed IoT devices in enhancing children's digital interest and cognitive development.
4. Disseminate research findings and promote the adoption of these IoT devices in education and at home settings.

Methodology:
To achieve our objectives, we plan to conduct surveys and focus groups with children aged 6-10 years old to identify their interests and preferences towards di
gital devices. Based on the results, we will then develop prototype IoT devices designed specifically for each age group. We aim to create fun, hands-on experi
ences that encourage learning while keeping children engaged.
We will evaluate the effectiveness of these devices through pilot testing conducted in schools and homes across different socioeconomic backgrounds. Participan
ts will have access to the IoT devices and regular assessments will be administered to gauge their level of interaction and interest. Findings from th
```

**Figure 18.** Sample output from using LLM for generation of research proposal.

Incorporating reinforcement learning principles [38] might enable systems to optimize their prompt construction strategies over time. By iteratively refining their queries based on user feedback, these machines could continually improve their performance without explicit guidance [23]. This adaptive nature would allow them to evolve along with the ever-changing landscape of online discourse. Of course, these examples only scratch what is possible when merging diverse AI techniques. In summary, combining NLP, text mining, and generative AI such as LLM and GPT can lead to exciting possibilities for enriching human tasks (e.g., literature reviews, seeking research ideas, etc.) in academic and professional publications [39]. By leveraging the strengths of these combinations, researchers and practitioners can develop innovative solutions beyond traditional boundaries. While challenges remain related to ensuring trustworthiness and usability, advances in these areas are promising for future progress in artificial intelligence, machine learning, and, more broadly, lifelong self-directed learning. As always, we need to stay tuned for updates and insights on this frontier area of research.

## 8. Conclusions

In our experiment with our developed AIR system, we successfully implemented Litstudy together with NLP under Mercury on an intranet/web platform. Three levels of analysis were successfully performed for the metadata of articles found in SpringerLink by students' searching with their key terms. The elementary level of analysis provided the demographic statistical analysis of the articles, which is helpful for the students to understand the background of the authors, institutes, regions, etc. of the articles. The intermediate level of analysis provided the articles' coupling network and citation network. These two network graphs provided innovative visualization of literature to the students, allowing them to trace ideas' origins and development trends by following the flow directions of citation. And to realize the similarity of ideas presented in the articles by observing the weights or thickness of the line connecting different nodes (i.e., the articles). The advanced level of analysis provided NLP and text mining functions, including topic modeling, frequent word analysis, word clouds, etc. These features could trigger the curiosity and creativity of the students in brainstorming research ideas and identifying gaps in current research. When used in an integrated way, these three levels of analysis can address the needs of students doing literature reviews at different stages or levels, ranging from understanding the background of the articles to creating new research ideas. The students can select the analysis they want to help them perform their literature review. Finally, we

also identified enhancement opportunities by integrating generative AI capabilities with the topic modeling functions we used in the advanced level of analysis to provide some forms of automatic generation of research ideas and draft research proposals. We did not aim to completely replace human participation in doing research. In reverse, we sought to integrate humans and machines in computer-aided (AI-backed) creativity to identify more research opportunities, achieving future human-in-the-loop integration with large language models [40].

**Author Contributions:** Conceptualization, G.K.W.W. and S.Y.K.L.; software, S.Y.K.L.; formal analysis, G.K.W.W.; investigation, G.K.W.W. and S.Y.K.L.; resources, G.K.W.W.; data curation, G.K.W.W.; writing—original draft preparation, S.Y.K.L.; writing—review and editing, G.K.W.W. and S.Y.K.L.; visualization, G.K.W.W. and S.Y.K.L.; supervision, G.K.W.W.; project administration, G.K.W.W.; funding acquisition, G.K.W.W. All authors have read and agreed to the published version of the manuscript.

**Funding:** This project is funded and supported by the Teaching Development Grant (Ref: 101002152) from the RGC/URC through the University of Hong Kong.

**Institutional Review Board Statement:** The study was conducted in accordance with the Declaration of Helsinki and approved by the Ethics Committee of the University of Hong Kong (HREC Reference Number: EA230330, Ethical Approval Period: From 31-08-2023 to 30-08-2025).

**Informed Consent Statement:** Informed consent was obtained from all subjects involved in the study.

**Data Availability Statement:** Data is unavailable due to privacy or ethical restrictions.

**Conflicts of Interest:** The authors declare no conflict of interest.

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
