# Peer review of "An Exploratory Study of Helping Undergraduate Students Solve Literature Review Problems Using Litstudy and NLP"

_education, doi:10.3390/educsci13100987_

Round 1

Reviewer 1 Report

1. I note that the research problem is not clear, and therefore the authors should clarify the real problem at the end of the introduction section.

2. All Figures are not clear. Thus, the authors should do it again and try to clarify more.

3. I suggest that authors should discuss the results more and show what their contributions are and what the new "develop system or what?"

5. I noticed the authors wrote a lot, but without references, and therefore new references must be added, especially in Section 2. Theoretical Background and Section 5 Discussion.

6. Authors should add a new section and state what the limitations are and what future research work is recommended.

Author Response

  1. I note that the research problem is not clear, and therefore the authors should clarify the real problem at the end of the introduction section.

Reply: Please refer to the reshuffled introduction section. The main objective of our research is to help the students in their literature review tasks by identifying research gaps. We developed a system to help the students in doing literature review.

  1. All Figures are not clear. Thus, the authors should do it again and try to clarify more.

Reply: All figures are enlarged in higher resolution.

  1. I suggest that authors should discuss the results more and show what their contributions are and what the new "develop system or what?"

Reply: A new section called “Contribution” is added.

  1. I noticed the authors wrote a lot, but without references, and therefore new references must be added, especially in Section 2. Theoretical Background and Section 5 Discussion.

Reply: Please refer to the added citations.

  1. Authors should add a new section and state what the limitations are and what future research work is recommended.

Reply: A new section called “Limitations” was added and the discussion section also contains the future research work.

Reviewer 2 Report

A really fascinating explanation of the technology was made here and I really hope to see it published. I have asked that some depth is added to help make the rationale and case study more detailed and stronger in illustrating the context.

Thank you. I enjoyed reading this work and look forward to seeing it published, but even more I hope to use the technology itself. It seems a potentially welcome tool.

Line 12 of abstract ‘any’ should read ‘and’ and it should be ‘reviews’ (plural)

Double space in line 13

Line 20 is boost, not boast

Line 22 ‘reviews’ (plural)

Line 35 refers to ‘this’ process and makes claims to success before these are outlined; I feel the process at least should first be outlined before using ‘this’, as the description up to now is not sufficiently detailed for the reader to know what you did. I would advise against early claims of success or for vast promise – make the case and let your reader judge at the end.

Explain NLP first time of use, rather than use initials. 

What happened on line 229? ‘specific’? and line 233? There appear to be strange glyphs in place of letters.

Spelling typo in Line 257 – educational

What is ’Ih’ in line 298?  A typo?

Line 334 be consistent with capitals or lower case used at start of bullet points

In Line 405 is ‘critical’ used as euphemism for ‘important’ or ‘popular’? It may be better to avoid the misleading term ‘critical’.  

Author Response

Line 12 of abstract ‘any’ should read ‘and’ and it should be ‘reviews’ (plural)

Reply: Solved.

Double space in line 13

Reply: Solved

Line 20 is boost, not boast

Reply: Solved

Line 22 ‘reviews’ (plural)

Reply: Solved

Line 35 refers to ‘this’ process and makes claims to success before these are outlined; I feel the process at least should first be outlined before using ‘this’, as the description up to now is not sufficiently detailed for the reader to know what you did. I would advise against early claims of success or for vast promise – make the case and let your reader judge at the end.

Reply: Solved. Please refer to line 57. The entire introduction section was reshuffled.

Explain NLP first time of use, rather than use initials. 

Reply: Solved. A new literature review section on NLP was also added. Please refer to line 203.

What happened on line 229? ‘specific’? and line 233? There appear to be strange glyphs in place of letters.

Reply: Solved. Please refer to lines 264 and 268

Spelling typo in Line 257 – educational

Reply: Solved. Please refer to line 294

What is ’Ih’ in line 298?  A typo?

Reply: Solved by rewriting the entire sentence. Please refer to line 341

Line 334 be consistent with capitals or lower case used at start of bullet points

Reply: Solved. Please refer to line 377.

In Line 405 is ‘critical’ used as euphemism for ‘important’ or ‘popular’? It may be better to avoid the misleading term ‘critical’. 

Reply: Solved. Please refer to line 457.

Round 2

Reviewer 1 Report

good luck